# Serum Folate, Red Blood Cell Folate, and Zinc Serum Levels Are Related with Gestational Weight Gain and Offspring’s Birth-Weight of Adolescent Mothers

**DOI:** 10.3390/nu16111632

**Published:** 2024-05-26

**Authors:** Reyna Sámano, Hugo Martínez-Rojano, Gabriela Chico-Barba, Ricardo Gamboa, Maricruz Tolentino, Alexa Xiomara Toledo-Barrera, Cristina Ramírez-González, María Eugenia Mendoza-Flores, María Hernández-Trejo, Estela Godínez-Martínez

**Affiliations:** 1Coordinación de Nutrición y Bioprogramación, Instituto Nacional de Perinatología, Secretaría de Salud, México City 11000, Mexico; gabyc3@gmail.com (G.C.-B.); cruz_tolentino@yahoo.com.mx (M.T.); crisra07@yahoo.com.mx (C.R.-G.); tina14mx@yahoo.com (M.E.M.-F.); eygodinez@hotmail.com (E.G.-M.); 2Sección de Estudios de Posgrado e Investigación, Escuela Superior de Medicina del Instituto Politécnico Nacional, México City 11340, Mexico; 3Coordinación de Medicina Laboral, Instituto de Diagnóstico y Referencia Epidemiológicos (InDRE) “Dr. Manuel Martínez Báez”, Secretaría de Salud, México City 01480, Mexico; 4Departamento de Fisiología, Instituto Nacional de Cardiología, México City 14080, Mexico; rgamboaa_2000@yahoo.com; 5Facultad de Nutrición, Universidad Autónoma del Estado de Morelos, Cuernavaca 62350, Mexico; alexa.xiomaratb@gmail.com; 6Departamento de Neurobiología del Desarrollo, Instituto Nacional de Perinatología, Secretaría de Salud, México City 11000, Mexico; maria.h.trejo72@gmail.com

**Keywords:** gestational weight gain, maternal diet, folic acid, folate, teenage pregnancy, newborn small for gestational age

## Abstract

Background: Gestational weight gain below or above the Institute of Medicine recommendations has been associated with adverse perinatal and neonatal outcomes. Very few studies have evaluated the association between serum and red blood cell folate concentrations and gestational weight gain in adolescents. Additionally, zinc deficiency during pregnancy has been associated with impaired immunity, prolonged labor, preterm and post-term birth, intrauterine growth restriction, low birth weight, and pregnancy-induced hypertension. Objective: The purpose of our study is to evaluate the association between serum concentrations of zinc, serum folate, and red blood cell folate, with the increase in gestational weight and the weight and length of the newborn in a group of adolescent mothers from Mexico City. Results: In our study, 406 adolescent-neonate dyads participated. The adolescents’ median age was 15.8 years old. The predominant socioeconomic level was middle-low (57.8%), single (57%), 89.9% were engaged in home activities, and 41.3% completed secondary education. Excessive gestational weight gain was observed in 36.7% of cases, while insufficient gestational weight gain was noted in 38.4%. Small for gestational age infants were observed in 20.9% of the sample. Low serum folate (OR 2.1, 95% CI 1.3–3.3), decreased red blood cell folate (OR 1.6, 95% CI 1.0–2.6), and reduced serum zinc concentrations (OR 3.3, 95% CI 2.1–5.2) were associated with insufficient gestational weight gain. Decreased serum zinc levels (OR 1.2, 95% CI 1.2–3.4) were linked to an increased probability of delivering a baby who is small for their gestational age. Conclusions: Low serum folate, red blood cell folate, and serum zinc concentrations were associated with gestational weight gain and having a small gestational age baby. Both excessive and insufficient gestational weight gain, as well as having a small gestational age baby, are frequent among adolescent mothers.

## 1. Introduction

Adolescent pregnancy is a public health concern among middle and low-income countries [1,2], with high costs to cover various perinatal complications [2] such as pre-eclampsia, anemia, low birth weight, and gestational weight gain, among others. Some complications can be derived from prepregnancy nutritional status and diet characteristics. The high prevalence of iron deficiency anemia occurs more frequently in pregnant women with the following characteristics: being a teenager, having low income, poor nutrition, and inadequate eating habits [3], which cause a deficiency of different nutrients essential for the pregnant woman and the fetus. Furthermore, approximately 66% of pregnant adolescents originating from Mexico have been shown to have inadequate gestational weight gain (GWG), resulting in adverse perinatal and neonatal outcomes [4,5,6].

It has been shown that inadequate gestational weight gain in pregnant adolescents is a consequence of poor eating habits, which are very common during adolescence. These habits often include diets rich in carbohydrates and saturated fats, misconceptions about nutrition during pregnancy, delayed or insufficient prenatal care, depression, economic constraints, and inadequate intake of nutritional supplements such as iron and folic acid [7,8,9].

Gestational weight gain serves as an indicator of nutritional status; however, studies have shown that energy intake in adolescent girls is not necessarily associated with GWG [10]. Pregnant women often fail to meet the recommended intake of both macronutrients and micronutrients [11], particularly adolescents and young women. Research indicates that only 30% of adolescent girls, both pregnant and non-pregnant, in Bangladesh, meet the recommended intake of zinc [12]. A study conducted by Singh A. et al. on 108 pregnant Latina adolescents demonstrated that less than 50% of them had adequate intake (excluding dietary supplements) of folate, vitamin A, vitamin E, iron, zinc, calcium, magnesium, and phosphorus [13]. Therefore, it is considered that the majority of pregnant adolescents do not reach the recommended levels of intake for macronutrients and micronutrients. This trend is consistent with findings from other studies, particularly in developing countries [7,10,14].

On the other hand, during pregnancy, serum and red blood cell folate, iron, and zinc concentrations decrease [15], along with other macronutrients and micronutrients. Therefore, it is essential for pregnant women to consume foods rich in these micronutrients and to take vitamin supplements at the beginning of pregnancy [14,16]. Deficiencies in these micronutrients have been associated with complications during pregnancy and childbirth, including premature birth, newborns small for gestational age (SGA) or with low birth weight, iron deficiency anemia, obstetric hemorrhage, postpartum hemorrhage, etc. [10,17,18].

Folate (vitamin B9) is a water-soluble vitamin that plays a vital role in deoxyribonucleic acid methylation, nucleic acids, and protein synthesis, making it a necessary nutrient in early pregnancy. Because the human body cannot synthesize folate, it must be obtained from foods or supplements. Sufficiency during pregnancy is important for normal fetal development [18]. Folate deficiency is associated with adverse outcomes including maternal anemia, neural tube defects, and birth defects [19,20]. However, because folic acid in natural foods is easily broken down during cooking and processing, the amount of folic acid alone is insufficient for pregnant women due to this loss. Therefore, folic acid supplements or multivitamins containing folic acid are recommended during pregnancy to prevent neural tube defects [21].

Zinc serves as a catalytic component for more than 200 enzymes and acts as a structural element in various nucleotides, proteins, and hormones. It plays essential roles in biochemical functions such as protein synthesis and nucleic acid metabolism, as well as cell division, gene expression, antioxidant defenses, wound healing, vision, and immune and neurological function [22]. Black et al. have reported that zinc deficiency contributes to approximately half a million maternal and infant deaths annually, primarily in developing countries [23]. Currently, there are no studies on the association of serum zinc concentrations or supplementation in pregnant adolescents, only in pregnant adults. In adults, it has been shown that zinc deficiency during pregnancy is associated with impaired immunity, prolonged labor, preterm and post-term birth, intrauterine growth restriction, low birth weight, and pregnancy-induced hypertension [24].

Pregnant adolescents are more likely to consume diets high in calories but low in micronutrients compared to adult pregnant women, which may result in complications during pregnancy. For example, in a study by Baker et al. [18] in the United Kingdom, which included 500 pregnant adolescents, iron and folate intake was found to be below the recommendations established by the United Kingdom and the United States. Moreover, 52% of the adolescents had iron deficiency anemia, and 30% had serum concentrations of 25-hydroxyvitamin D below 25 nmol/L, compromising both adequate gestational weight gain and the weight and development of the newborn [7,25]. Folate and zinc play a critical role in fetal growth and development, as demonstrated in adult pregnant women. However, in the case of pregnant adolescents, many of whom have not yet completed their growth and development, it is not known whether deficiencies in these micronutrients have a greater impact on both the adolescent and her children. Furthermore, numerous observational studies in adult pregnant women suggest a potential benefit of adequate maternal folate concentration on newborn weight and the duration of gestation [26,27]. Therefore, the purpose of our study is to evaluate the association between serum concentrations of zinc, serum folate, and folate in red blood cells, with GWG, and the weight and length of the newborn in a group of adolescent mothers from Mexico City.

## 2. Methods

### 2.1. Study Design

A prospective cohort design was implemented with 406 pairs of adolescent mothers and newborns. The pregnant adolescents were clinically and metabolically healthy, received prenatal care, and delivered at the National Institute of Perinatology (INPer). Pregnant adolescents were invited to participate in our study while attending outpatient clinic appointments. As is common in Mexico, all pregnant women were recommended to take a supplement containing iron, folic acid, and other micronutrients. This recommendation is mandatory in every healthcare center at all levels of medical care [28].

During the years 2016–2021, information was collected. All pregnant adolescents with no history of chronic diseases, who began their prenatal care at the INPer and delivered in the obstetrics unit of the INPer, were included in the study. Adolescents whose pregnancies resulted from rape, or who had pre-existing metabolic conditions such as type 1 diabetes, psychiatric, autoimmune, or cardiac diseases, high blood pressure, kidney failure, hypothyroidism, or whose offspring had congenital defects, were excluded from the study. Adolescents who experienced fetal death or withdrew their informed consent or assent were also excluded from our analysis.

### 2.2. Serum Folate, Red Blood Cell Folate, and Serum Zinc Determination

The teenager’s blood sample was obtained in the morning, following an 8-hour fast. The sample was centrifuged at 3000 revolutions per minute for 10 min to obtain a serum, which was then stored at −70 °C until determination of zinc and folate levels. Serum and erythrocyte folate were assessed using the INMULITE-1000 kit and the chemiluminescence immunometric assay method. After determination, cut-off points were assigned to define deficiency (<3 ng/mL), marginal concentration (3–5.9 ng/dL), normal concentration (6–20 ng/dL), and elevated concentration (>20 ng/dL) according to WHO guidelines [29,30]. However, due to very few cases with insufficient levels in our sample, we classified them into tertiles.

On the other hand, atomic absorption spectrophotometry was conducted using PerkinElmer Analyst 400 equipment to determine serum zinc concentrations. The data were expressed in μg/L [31] and subsequently classified into tertiles or medians.

Hemoglobin levels were determined using a Beckman Coulter device, with results expressed in g/dL. Maternal anemia status was classified as hemoglobin <12.5 g/dL or ≥12.5 g/dL. The cut-off point of <12.5 g/dL of hemoglobin was chosen because Mexico City is located 2300 m above sea level [32].

### 2.3. Anthropometric Evaluation

All anthropometric measurements were carried out by qualified personnel in accordance with the standardization of anthropometric measures [33]. Body weight was assessed both at the beginning of the study and at the end of pregnancy. Pregestational weight was self-reported [34,35], and maximum gestational weight was measured using a digital scale (TANITA, Tokyo, Japan, model BWB-800, accuracy 0.10 kg) one week before delivery, while height was measured using a metal stadiometer (SECA brand, Hamburg, Germany, model 208, with a precision of 0.1 cm).

Body mass index (BMI) (in kg/m^2^) was calculated by dividing prepregnancy (observed) or first-trimester (observed or imputed) weight in kilograms by the square of height in meters. As the participants were adolescents, BMI classification was obtained using AnthroPlus^®^ (World Health Organization, Geneva, Switzerland, 2006) according to percentiles: underweight <3rd percentile, normal weight 3rd to <85th percentile, overweight 85th to 97th percentile, and obesity ≥97th percentile [36,37].

### 2.4. Gestational Weight Gain

We determined the total gestational weight gain by subtracting the prepregnancy weight from the final weight in kilograms. First, for each adolescent, we calculated the gestational weight gain at the time of the last weight measurement during pregnancy by subtracting the prepregnancy or first-trimester weight from the last available weight measurement during pregnancy. Second, following the 2009 Institute of Medicine (IOM) recommendation, we estimated the expected weight gain for each adolescent at the time of her last observed weight measurement. Then, based on the recommendations of the IOM [38] of the United States of America, we classified the gestational weight gain according to each category of prepregnancy BMI (pBMI). The expected weight gain was calculated using the following equation [39]:Expected weight gain = recommended weight gain for the first trimester + ((gestational age final − 13.86 weeks) × (recommended weight gain rate in second and third trimesters)).

The recommended GWG rate for the first trimester varied according to prepregnancy BMI (pBMI): 2 kg for low and normal weight, 1 kg for overweight, and 0.5 kg for obesity. For adolescents in the second and third trimesters, these pBMI-based figures were as follows: 0.51 kg for low weight, 0.42 kg for normal weight, 0.28 kg for overweight, and 0.22 kg per week for obesity [38].

Subsequently, the percentage of GWG adequacy was calculated by dividing the observed GWG at the time of the last weight measurement by the expected GWG for that week of gestation according to IOM recommendations, then multiplying by 100. We categorized the GWG percentage as follows: inadequate (<90%), adequate (90 to <125%), and excessive (≥125%).

### 2.5. Neonatal Outcomes

Gestational age was determined using the Capurro method and recorded in weeks and days. If the gestational age was under 36.6 weeks, it was classified as preterm; gestational ages between 37 and 42 weeks were identified as term deliveries.

Weight (measured with SECA 374, model “Baby and Mommy”; accuracy 0.1 g) and length (measured with stadiometer SECA 416; accuracy 0.1 cm) at birth were recorded. Low birth weight (LBW) was defined as <2500 g and SGA was defined as birth weight below the 10th percentile for specific gestational age and sex, while appropriate for gestational age (AGA) was between the 10th and 90th percentiles, and large for gestational age (LGA) was above the 90th percentile, according to World Health Organization (WHO) criteria using INTERGROWTH- 21 [40].

### 2.6. Dietary and Nutrient Intake

Three 24-hour dietary recalls were conducted, with two recorded on non-consecutive weekdays and one on a weekend day. Trained personnel administered the 24-hour recalls. Photographs of commonly consumed foods in Mexico were provided to aid in portion size estimation; these were given to participants during the first 24-h dietary recall and retained for subsequent interviews. Nutrient and energy intake were estimated using NutriKcal^®^ software. Subsequently, the mean energy intake in kilocalories (kcal) was calculated. To assess participants’ energy intake adequacy, we referred to the Institute of Medicine (IOM) guidelines (2005) and categorized energy intake as insufficient (<80%), adequate (80–119%), or excessive (>120%).

The contribution of folate, zinc, iron, legumes, sugar, cereals, milk and derivatives, carbohydrates, proteins, and lipids to total energy consumption was estimated. We used the recommendations of the IOM as a reference to categorize the distribution of macronutrient energy contribution [38]. Folic acid supplementation in Mexico was accounted for in the dietary intake, with a recommended daily dose of 400 micrograms (equivalent to 680 micrograms daily as dietary folate equivalents) [41].

### 2.7. Other Variables

Other variables were obtained from clinical records, including gestational age in weeks and type of delivery (cesarean section or vaginal delivery). We recorded sociodemographic information, including chronological age, education level (elementary, middle school, and high school), and marital status (single, married, and cohabiting).

Finally, to determine the socioeconomic level, we utilized a questionnaire validated for the Mexican population. The resulting categories within our sample were middle, low-middle, and low [42].

### 2.8. Ethical Aspects

Our research received approval from the institutional ethics, research, and biosafety committees, with registration numbers 212250-49541-INPer and 2017-2-101-INPer. The ethical approval dates were 2012 and 2017. All adolescents and their guardians were informed about the study’s objectives and procedures. Confidentiality was ensured by assigning an ID number to each participant during data collection and analysis. Written informed consent and assent were obtained from both adolescents and guardians. We adhered to the guidelines of the Helsinki Declaration. Additionally, all adolescents received medical attention at INPer.

### 2.9. Statistical Analysis

We conducted univariate analysis to describe the general characteristics of our sample according to the type of variable. Tertiles of serum and red blood cell folate were compared with gestational weight gain and birth weight categories using the chi-square test. Additionally, we compared medians for continuous variables using the Kruskal–Wallis test, and post-hoc tests were performed using the Mann–Whitney test. Bivariate correlations were calculated using the Spearman Rho test.

To address the main objective, we converted gestational weight gain, birth weight, serum folate, red blood cell folate, and serum zinc into dummy variables. Insufficient or excessive gestational weight gain was treated as an outcome, with adequate gestational weight gain as the reference. For birth weight, SGA was the outcome, with adequate birth weight as the reference. Serum folate, red blood cell folate, and zinc were categorized based on the median of the sample, with values below the median considered as risk exposure variables.

Subsequently, logistic regression models were performed to analyze the association between the outcomes and exposure variables. The models were adjusted for sociodemographic variables, energy intake, dietary nutrients, food intake, gestational age, and trimester of beginning antenatal care. Statistical analysis was conducted using IBM SPSS Version 23 (IBM Inc., Armonk, NY, USA).

## 3. Results

In our study, 406 adolescent-neonate dyads participated. Of the adolescent mothers, 57% were single, while the remaining cohabited with a partner. A total of 89.9% were engaged in home activities, and 10% were students. The predominant socioeconomic status was middle–low, accounting for 57.8%, followed by low at 31%, with the remainder classified as high–middle socioeconomic status. Regarding educational level, 41.3% had completed secondary education, 33.4% had completed high school, and 25.2% had concluded with an elementary grade. Table 1 presents the general characteristics of our participants.

Adolescents with insufficient GWG had lower serum folate, red blood cell folate, serum zinc, and hemoglobin compared with adequate and excessive GWG. Similarly, SGA neonates had lower serum folate, red blood cell folate, and serum zinc compared with adequate and large for gestational age, as can be observed in Table 2. Dietetic variables and the serving of food did not have any difference.

Figure 1 depicts correlations between serum folate, red blood cell folate, and serum zinc with GWG. However, serum folate did not correlate with birth weight. Not all variables were found to correlate with both GWG and birth weight.

It was observed that the low tertile (tertile 1) of serum folate, red blood cell folate, and serum zinc had a higher frequency of insufficient GWG, as shown in Figure 2. Additionally, below-median levels of red blood cell folate and serum zinc were associated with a higher percentage of SGA neonates, as depicted in Figure 3.

We observed that serum folate, red blood cell folate, serum zinc, and hemoglobin levels increased the risk of insufficient GWG. In contrast, factors such as being overweight or obese, being under 15 years old, consuming less than 25 g of fiber, and inadequate fruit intake were associated with a 70% decrease in the risk of insufficient GWG. Elevated serum folate and serum zinc levels, pregestational overweightness or obesity, low fiber intake, and excessive intake of fats and oils increased the risk of excessive GWG. Additionally, below-median concentrations of zinc were associated with an increased risk of neonates being small for gestational age, while being overweight decreased the risk by 50%, as observed in Table 3.

In our sample, prematurity was associated with a 76% increased risk when red blood cell folate levels were below the median (832 ng/dL), with a beta coefficient of 1.76, a 95% confidence interval of 0.96–3.22, and a *p*-value of 0.067 in the binary logistic regression model.

## 4. Discussion

In our study, we found that insufficient concentrations of serum folate, red blood cell folate, and serum zinc were associated with inadequate GWG and SGA newborns. However, dietary factors such as fiber, fruits, and fats and oils were only partially associated with gestational weight gain or having a small-for-gestational-age newborn.

There are several plausible mechanisms through which folic acid, folates, and zinc may influence GWG. Firstly, folic acid and zinc have been shown to reduce the risk of infections and morbidities during pregnancy [43]. Additionally, folic acid aids in improving immune function and reducing the risks of pre-eclampsia and eclampsia during pregnancy [44]. Secondly, folic acid directly contributes to fetal development and growth, thereby promoting higher GWG [45]. For instance, folic acid is involved in energy and protein metabolism, as well as in the synthesis of deoxyribonucleic acid (DNA), ribonucleic acid (RNA), and cell division [46]. Folic acid deficiency has been linked to adverse pregnancy outcomes, including low birth weight and preterm birth [47].

### 4.1. Gestational Weight Gain

It is noteworthy that the frequency of normal weight before pregnancy was 70%, despite only 25% of all women achieving adequate weight gain, with 35% experiencing excessive GWG. This distribution differs from that of other study groups in the United States, where nearly 60% had excessive GWG and 16% had insufficient GWG. However, the proportion achieving adequate weight gain was similar. This raises concern because adolescent mothers may be at risk of retaining excess weight postpartum, as documented in a study where adolescents who gained above the recommended levels retained 9.5 kg [48]. Given the high prevalence of obesity and overweightness in the Mexican female population of reproductive age [49], adolescents who experience pregnancy may be at increased risk of gaining excessive weight. While we observed excess GWG, we found a low frequency of folate insufficiency and an increased risk of insufficient GWG or SGA infants among their offspring.

### 4.2. Folate and Gestational Weight Gain

The results of our study revealed a median folate level of 13.2 ng/dL, with only 4.8% of participants showing insufficiency in folate. As for red blood cell folate, 98.2% of the participants had normal levels, which is higher compared to other samples of pregnant adolescents [50] from the United Kingdom.

We found that low serum folate levels were associated with insufficient GWG. This association can be explained by the role of one-carbon metabolism, which provides methyl groups crucial for the development and growth of the fetus [51,52]. In adult women, folate deficiency during pregnancy is associated with low birth weight [53,54]. Adolescents are known to have inadequate intake of folate and other B-group vitamins [55]. We believe that folic acid supplementation during antenatal care has been beneficial. Despite the possibility of poor diet quality in our sample concerning B-group vitamins, their folate levels met normal parameters [29]. Supplementation during pregnancy likely played a role in preventing adverse perinatal outcomes [56,57], as many supplements provide 98% of the recommended folate intake for pregnancy [58]. At our institute, all women are required to be supplemented with folic acid and iron according to legal regulations [28].

In our research, we reported that tertile 1 of serum and red blood cell folates was associated with a higher frequency of insufficient GWG. This could be explained by the folate status in serum and red blood cells. Both types of folates have been associated with growth, although their impact on adolescent mothers remains uncertain. Adolescents comprise a group that has not completed their growth and competes with their fetuses for nutrients [50].

On the other hand, Jones RL et al. demonstrated in a group of 368 pregnant adolescents that the average concentrations of folate in red blood cells were 15% higher in adolescents who experienced growth during pregnancy and consequently showed an adequate increase in gestational weight compared to those who did not grow (*p* < 0.05). Additionally, a similar trend was observed for serum folate and vitamin B12, although these differences did not reach statistical significance after adjustment for confounding variables [50].

There is also evidence that maternal folate concentrations are higher in growing adolescent girls, although this difference was only significant for erythrocyte folate. However, as folate is an essential component of growth and development during all stages of life, it is biologically plausible that insufficiency during pregnancy could affect maternal and fetal growth rates [59].

### 4.3. Folate, Small Newborns for Gestational Age, and Duration of Pregnancy

In our study, we reported a higher frequency of SGA newborns at 20.9%, compared to 17.1% in a group of adolescents from the United Kingdom [50], as well as an average of 10% in different national surveys of pregnant adolescents conducted in Brazil [60]. Our results are consistent with those reported by Jones RL et al. [50] in a cohort of 368 mother-infant pairs, where an association between poor maternal folate status and the birth of small-for-gestational-age babies was identified.

Regarding preterm birth, we did not observe an association between dietary folate intake and serum and erythrocyte folate concentrations with the duration of pregnancy, as demonstrated by a systematic review with a meta-analysis by Fekete et al. [27], who did not find a significant effect of folate supplementation on gestation duration in the intervention groups compared to the placebo groups (*p* = 0.77). On the contrary, Siega-Riz et al. [61] conducted a prospective study with more than 2000 pregnant women, which demonstrated that low serum folate concentrations are associated with almost double the risk of preterm birth. Similarly, the study by Olapeju B et al. [53], which reported on a group of 2313 mother–child dyads from the Boston Birth Cohort, demonstrated that adequate concentrations of plasma folate were associated with a lower risk of preterm birth (aOR: 0.74; 95% CI: 0.56–0.97).

This discrepancy may be due to the lack of methodological information in studies published several years ago. For example, laboratory parameters of the included pregnant women and other potential confounding factors, such as smoking, alcohol consumption, maternal BMI, or the sex of the baby, were generally poorly described. The substantial risk of bias increases the uncertainty of the results and may lead to an overestimation or underestimation of the true effect of serum and erythrocyte folate concentrations, as well as dietary intake.

### 4.4. Zinc, Gestational Weight Gain, and Small Newborns for Gestational Age

Insufficient serum zinc concentrations in adolescents are associated with inadequate GWG and SGA newborns. Our results are partially consistent with those reported by Ota et al. [24], who found that zinc deficiency during pregnancy in adult women was associated with impaired immunity, pregnancy-induced hypertension, prolonged labor, preterm and post-term birth, delayed intrauterine growth, and babies with low birth weight. To date, the benefits of zinc supplementation during pregnancy have not been conclusively demonstrated, and since zinc deficiency often reflects poor diet, strategies aimed at improving maternal nutrition are likely to yield more tangible health benefits than relying exclusively on zinc supplements [62].

### 4.5. Other Findings

Our findings indicate that fiber, fruits, fats, and oils were also associated with GWG. A balanced diet rich in fiber derived from fruits, vegetables, and legumes correlates with favorable outcomes, likely owing to their inherent nutritional properties [63]. Caution should be taken with those women with low fruit consumption, as they may be at low risk of insufficient GWG but could be at higher risk of malnutrition due to the loss of the beneficial effects of vitamins, antioxidants, and other anti-inflammatory components found in fruits. On the contrary, a high intake of fats and oils increases the risk of excessive GWG; however, it is also associated with metabolic health complications for the mother and her offspring over the short and long term. This risk is exacerbated by the prevalence of high-fat content in many ultra-processed foods [64,65], which are known to have detrimental effects on maternal health during pregnancy. The recommendation is to have a diverse diet to prevent adverse events during pregnancy in adolescents.

### 4.6. Strengths and Weaknesses

Studies involving pregnant adolescents can be challenging due to their transient lifestyles and low attendance at prenatal clinics, which introduce difficulties for participant retention. Despite these challenges, this cohort study has investigated the impact of gestational weight gain, incorporating anthropometric measurements, nutritional assessments, evaluations of serum and erythrocyte folate, as well as serum zinc levels.

## 5. Conclusions

We found that insufficient concentrations of serum folate, red blood cell folate, and serum zinc were associated with inadequate gestational weight gain and an SGA newborn. However, diet and nutrient intake were only partially associated with GWG. Most previously published studies were conducted in pregnant adults and focused on the effect of folates on birth outcomes rather than GWG. This is because birth outcomes have long been prioritized over maternal outcomes. Future research efforts should be directed toward studying the determinants and consequences of maternal outcomes during adolescence.

## Figures and Tables

**Figure 1 nutrients-16-01632-f001:**
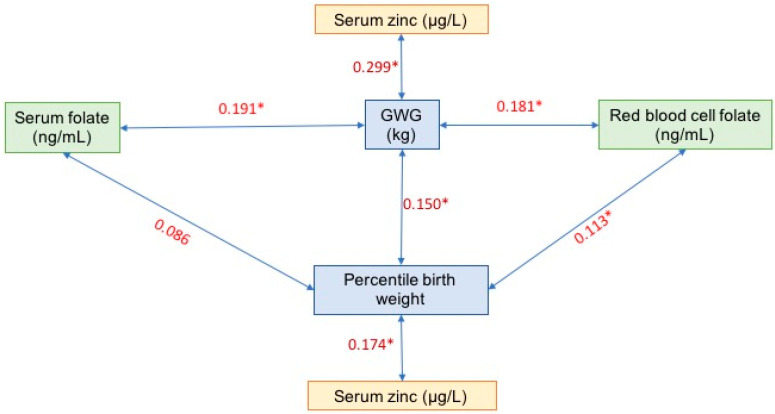
Rho of Spearman correlation of GWG and percentile birth weight; * *p* ≤ 0.05.

**Figure 2 nutrients-16-01632-f002:**
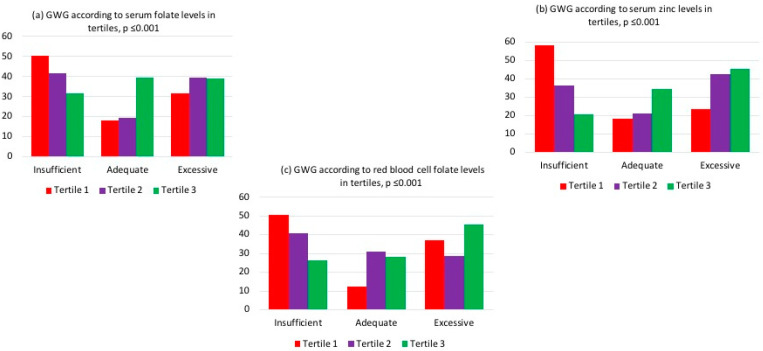
Gestational weight gain according to (**a**) serum folate, (**b**) red blood cell folate, and (**c**) serum zinc levels in tertiles, with *p*-values calculated using Pearson’s chi-squared test.

**Figure 3 nutrients-16-01632-f003:**
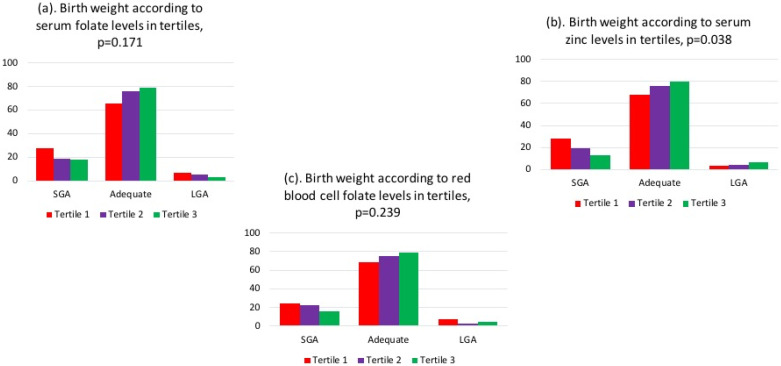
Birth weight according to (**a**) serum folate, (**b**) red blood cell folate, and (**c**) serum zinc levels in tertiles, with *p*-values calculated using Pearson’s chi-squared test.

**Table 1 nutrients-16-01632-t001:** General characteristics of participants, n = 406.

Maternal Variables	Mean ± SD	Interval
Age (years)	15.8 ± 1.3	12–19
Gynecological (years)	4.3 ± 1.6	1–10
Beginning of antenatal care (weeks)	27 ± 6	11–37
Delivery ^(a)^	Cesarean-section	173 (42.7)
	Natural	233 (57.3)
Height (cm)	155.9 ± 5.5	139.4–176
Prepregnancy BMI	21.5 ± 3.6	13.5–39.1
Prepregnancy BMI (percentile)	52.5 ± 10.1	1–99
Prepregnancy BMI ^(a)^	Low weight	36 (8.9)
	Healthy	286 (70.4)
	Overweight	62 (15.3)
	Obesity	22 (5.4)
Prepregnancy weight (kg)	52.5 ± 10	28–100
Adequacy of gestational weight gain (%)	114 ± 66	−128–359
Adequacy of gestational weight gain (kg)	12.7 ± 6	−7.7–35.5
Gestational weight gain ^(a)^	Inadequate	156 (38.4)
	Recommendable	101 (24.9)
	Excessive	149 (36.7)
Hemoglobin (g/dL)	12.7 (11.8–13.3)	9.2–15.9
Serum folate (ng/dL)	13.2 (9.4–0.3)	3.93–94.4
Red blood cell folate (ng/dL)	370 (260–490)	63.5–1684
Serum zinc (μg/L)	665 (575–798)	164–1481
Perinatal outcomes		
Birth weight (g)	2929 ± 477.7	1030–4105
Birth weight for gestational age ^(a)^	Small	85 (20.9)
	Adequate	302 (74.4)
	Large	19 (4.7)
Length (cm)	48.8 ± 2.7	31–53
Gestational age (weeks)	39 (37.6–40.0)	26.6–42
	Preterm	57 (13.8)
	Term	349 (86.2)
Sex (gender) ^(a)^	Women	189 (46.7)
	Men	217 (53.3)

^(a)^ Frequency (%); SD: Standard deviation.

**Table 2 nutrients-16-01632-t002:** Serum folate, red blood cell folate, serum zinc, dietetic nutrients, and food intake according to GWG and birth weight.

	Gestational Weight Gain			Birth Weight		
	Insufficient, n = 156	Adequate n = 101	Excessive, n = 149	Small, n = 85	Adequate n = 302	Large, n = 19
Micronutrients						
Serum folate (ng/mL)	11 (8–16) ^1–2,1–3^	17 (11–29)	14 (10–22)	14 (9–19)	13 (10–23)	12 (8–16)
Red blood cell folate (ng/mL)	327 (217–429) ^1–2,1–3^	389 (323–530)	388 (277–548) *	349 (278–453) ^1–2^	378 (278–501)	290 (245–441)
Serum zinc (μg/L)	602 (533–694) ^1–2,1–3^	715 (611–845)	711 (632–870) *	616 (514–713) ^1–2,1–3^	679 (589–801)	732 (610–884) *
Hemoglobin (g/dL)	12.1 (11.2–13.0) ^1–2,1–3^	13.3 (12.5–13.7)	12.7 (11.9–13.1)	12.6 (11.6–13.2)	12.8 (11.9–13.4)	12.4 (11.4–12.9)
Dietetic variables						
Energy (kcal)	1821 (1491–2188)	1803 (1524–2283)	1776 (1402–2100)	1798 (1456–2191)	1794 (1450–2166)	1940 (1697–2252)
Fiber (g)	13 (9–19)	12 (8–16)	12 (8–15) ^1–3^	13 (8–19)	12 (9–17)	11 (7–15)
Folate (µg)	854 (771–965)	822 (766–950)	840 (767–921)	977 (771–997)	838 (767–920)	818 (786–874)
Thiamine B1 (mg)	1.1 (0.8–1.5)	1.0 (0.8–1.5) ^1–2^	1.1 (0.8–1.4)	1.2 (0.8–1.7)	1.0 (0.8–1.4)	1.1 (0.9–1.3)
Riboflavin B2 (mg)	1.4 (1.0–1.9)	1.3 (0.8–1.8)	1.4 (0.9–1.8)	1.4 (0.9–1.8)	1.3 (0.9–1.8)	1.3 (1.0–1.7)
Pyridoxine B6 (mg)	1.2(0.8–1.8)	1.1 (0.7–1.8)	1.0 (0.7–1.4)	1.3 (0.7–1.8)	1.1 (0.7–1.6)	0.9 (0.8–1.3)
Cyanocobalamin B12 (µg)	2.4 (1.4–3.5)	2.6 (1.4–3.6)	2.2 (1.3–3.5)	2.5 (1.5–3.7)	2.3 (1.3–3.5)	1.5 (1.2–2.6)
Pantothenic acid (mg)	2.4 (1.8–3.4)	2.4 (1.3–4.0)	2.3 (1.4–3.3)	2.7 (1.5–3.9)	2.3 (1.6–3.2)	2.5 (1.7–2.8
Niacin (mg)	14 (9–19)	14 (10–18)	12 (9–15)	15 (10–20)	13 (9–18)	12 (10–18)
Iron (mg)	12 (9–19)	12 (9–17)	12 (9–15)	12 (9–20)	12 (9–16)	11 (9–14)
Zinc (mg)	5 (4–8)	6 (4–8)	5 (4–8)	5 (4–8)	5 (4–8)	4 (4–6)
Serving of food						
Fruits	1.6 (0.1–3.2)	1.2 (0.3–2.5)	1.1 (0.1–2.7)	1.4 (0.0–3.4)	1.4 (0.2–2.8)	0.7 (0.0–1.2)
Vegetables	1.7 (0.8–3.1) ^1–3,2–3^	1.8 (0.7–2.9)	1.2 (0.4–2.8)	1.2 (0.5–3.1)	1.5 (0.6–2.9)	1.4 (0.6–2.2)
Legumes	0.0 (0–0.5)	0.0 (0–0.5)	0.0 (0–0.5)	0.0 (0.0–0.7)	0.0 (0.0–0.5)	0.0 (0.0–0.6)
Sugar	3.3 (1.2–6.1) ^1–3,2–3^	2.8 (1.2–4.4)	4 (2.1–6.0)	3.7 (2.1–6.2)	3.3 (1.5–6.0)	4.5 (1.3–7.0)
Animal-sourced foods	5 (3.5–7.1)	5.1 (3.4–6.6)	4.9 (2.8–6.6)	5.1 (3.8–6.6)	4.8 (3.0–6.8)	4.2 (3.6–7.0)
Cereal	8 (6–11)	8 (7–10)	8 (5–11)	8 (6–11)	8.3 (6–11)	10 (6–11)
Milk and derivate	1.0 (0.5–1.9)	1 (0–2)	1 (0–2)	1.0 (0.0–1.9)	1.0 (0.1–1.9)	1.0 (0.6–1.6)
Fat and oils	3.9 (1.5–6.5) ^1–3,2–3^	3.4 (1.5–7.7)	5.3 (3.0–7.0)	4.0 (1.5–6.6)	4.2 (2.0–7.0) ^2–3^	5.1 (4.1–9.0)
Sweetened beverages	1 (0–3) ^1–3,2–3^	1 (0–3)	2 (2–4)	2 (1–4)	2 (0–3)	2 (1–3)

* *p* ≤ 0.001 by Kruskal–Wallis. ^1–2^ Insufficient vs. adequate 0.001, ^1–3^ insufficient vs. excessive, ^2–3^ adequate vs. excessive. U-Mann–Whitney post-hoc test *p* ≤ 0.050.

**Table 3 nutrients-16-01632-t003:** Associated variables with gestational weight gain and birth weight. Binary logistic regression.

	Gestational Weight Gain		Birth Weight
	Insufficient, n = 156	Excessive, n = 149	SGA, n = 85
	OR (CI 95%)	OR (CI 95%)	OR (CI 95%)
Serum folate below the median	2.1 (1.3–3.3)	0.6 (0.3–0.9)	0.9 (0.5–1.5)
Red blood cell folate (ng/mL) below median	1.6 (1.0–2.6)	0.7 (0.4–1.1)	1.6 (0.9–2.7)
Serum zinc (μg/L) below median	3.3 (2.1–5.2)	0.5 (0.3–0.7)	1.2 (1.2–3.4)
<15 years old	0.5 (0.3–0.8)	1.2 (0.7–1.9)	1.2 (0.7–2.0)
Hemoglobin < 12.5 (g/dL)	1.9 (1.2–3.1)	1.0 (0.6–1.6)	0.9 (0.5–1.6)
pBMI overweight/obese	0.3 (0.2–0.7)	4.9 (2.1–8.5)	0.5 (0.2–0.9)
Dietetic variables			
Fiber < 25 g	0.3 (0.1–0.6)	2.2 (1.0–4.9)	0.5 (0.2–1.0)
Fruits inadequate intake	0.3 (0.2–0.7)	1.3 (0.6–2.8)	0.8 (0.4–1.7)
Legumes inadequate intake	1.4 (0.4–4.4)	0.4 (0.1–1.4)	2.7 (0.6–11.3)
Fat and oils	0.6 (0.4–1.1)	2.1 (1.3–3.4)	0.8 (0.5–1.3)

Adjusted by sociodemographic variables, energy intake, dietary nutrients, food intake, and gestational age; SGA: small for gestational age; OR: odds ratio; CI: confidence interval.

## Data Availability

The data presented in this study are available from the corresponding author upon reasonable request.

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
