# Peer review of "Serum Folate, Red Blood Cell Folate, and Zinc Serum Levels Are Related with Gestational Weight Gain and Offspring’s Birth-Weight of Adolescent Mothers"

_nutrients, 2024, doi:10.3390/nu16111632_

Round 1

Reviewer 1 Report

Comments and Suggestions for Authors

Dear editors, thank you very much for the opportunity to review this article.
Each pregnancy in an adolescent patient should be treated with increased caution and managed in an experienced center.
Avoid using abbreviations in the abstract.
Very good structure of the article.
The material and methods are clearly presented.
Results presented transparently
Unfortunately, the literature is somewhat outdated, citing articles from 30 years ago seems inappropriate.
There are numerous articles in the current literature that concern changes in GWG during pregnancy, for example in patients after bariatric surgery - it is worth finding out.

Author Response

Reviewer 1

We appreciate the time taken to review our manuscript.

  1. Avoid using abbreviations in the abstract.

Thank you very much for the observation. The abbreviations in the abstract have been removed.

  1. Unfortunately, the literature is somewhat outdated, citing articles from 30 years ago seems inappropriate.

The references have been updated.

  1. There are numerous articles in the current literature that concern changes in GWG during pregnancy, for example in patients after bariatric surgery - it is worth finding out.

Thank you for the observation. You are right, there are numerous articles that refer to changes in gestational weight gain during pregnancy, but very few refer to gestational weight gain in pregnant adolescents.

Reviewer 2 Report

Comments and Suggestions for Authors

The study "Serum Folate, Red Blood Cell Folate, and Zinc Serum Levels are Related with Gestational Weight Gain and Offspring’s Birth-Weight of Adolescent Mothers" addresses a crucial topic focusing on a specific demographic. Given the increased health risks associated with adolescent pregnancy, it is both timely and relevant. The manuscript is generally well-written, with a logical flow from the introduction to the conclusion.

The authors should be very careful in interpreting the results of other studies. Too many discrepancies are found when quoting other studies.

Lines 75-79: The authors mention that references 12 and 13 are from India. The 12th reference is from Bangladesh, not India, and the 13th reference does not even have any connection with adolescent pregnancy. It is too unfortunate to see the misquoting and providing the wrong information.

Lines 114-116: again, the same issue. The 23rd reference does not have a concern with adolescent pregnancy in India; similarly, the 25th reference talks about Bangladesh, not India.

The authors should try to understand a nation's policy and demographics before making such a generalized statement.

Benefiting the prospective cohort design provides robustness to the findings; I recommend a thorough revision of this manuscript and proper citation of the references in the revised version.

Comments on the Quality of English Language

Minor editing should be done.

Author Response

Reviewer 2

We appreciate the time taken to review our manuscript.

  1. Lines 75-79: The authors mention that references 12 and 13 are from India. The 12th reference is from Bangladesh, not India, and the 13th reference does not even have any connection with adolescent pregnancy. It is too unfortunate to see the misquoting and providing the wrong information.

Regarding reference 12, the error in the location of the country where the research was conducted has been corrected, and reference 13 has been updated

  1. Lines 114-116: again, the same issue. The 23rd reference does not have a concern with adolescent pregnancy in India; similarly, the 25th reference talks about Bangladesh, not India.

The change to reference 23 has been made, and the error in reference 25 concerning the location of the country where the study was conducted in the manuscript has been corrected.

  1. Benefiting the prospective cohort design provides robustness to the findings; I recommend a thorough revision of this manuscript and proper citation of the references in the revised version.

A thorough review of the manuscript was conducted concerning the references used to support our research, as well as their interpretation in the manuscript.